# Analysis of the Spatial-Temporal Distribution Characteristics of Climate and Its Impact on Winter Wheat Production in Shanxi Province, China, 1964–2018

**DOI:** 10.3390/plants13050706

**Published:** 2024-03-01

**Authors:** Donglin Wang, Mengjing Guo, Xuefang Feng, Yuzhong Zhang, Qinge Dong, Yi Li, Xuewen Gong, Jiankun Ge, Feng Wu, Hao Feng

**Affiliations:** 1College of Water Conservancy, North China University of Water Resources and Electric Power, Zhengzhou 450046, Chinagongxuewen@ncwu.edu.cn (X.G.); wufeng@ncwu.edu.cn (F.W.); 2Institute of Soil and Water Conservation, Chinese Academy of Sciences and Ministry of Water Resources, Yangling, Xianyang 712100, China; 3Henan Key Laboratory of Water-Saving Agriculture, Zhengzhou 450046, China

**Keywords:** climate change, actual yield, climatic yield, variation coefficient, correlation analysis

## Abstract

The possible influence of global climate changes on agricultural production is becoming increasingly significant, necessitating greater attention to improving agricultural production in response to temperature rises and precipitation variability. As one of the main winter wheat-producing areas in China, the temporal and spatial distribution characteristics of precipitation, accumulated temperature, and actual yield and climatic yield of winter wheat during the growing period in Shanxi Province were analysed in detail. With the utilisation of daily meteorological data collected from 12 meteorological stations in Shanxi Province in 1964–2018, our study analysed the change in winter wheat yield with climate change using GIS combined with wavelet analysis. The results show the following: (1) Accumulated temperature and precipitation are the two most important limiting factors among the main physical factors that impact yield. Based on the analysis of the ArcGIS geographical detector, the correlation between the actual yield of winter wheat and the precipitation during the growth period was the highest, reaching 0.469, and the meteorological yield and accumulated temperature during this period also reached its peak value of 0.376. (2) The regions with more suitable precipitation and accumulated temperature during the growth period of winter wheat in the study area had relatively high actual winter wheat yields. Overall, the average actual yield of the entire region showed a significant increasing trend over time, with an upward trend of 47.827 kg ha^−1^ yr^−1^. (3) The variation coefficient of winter wheat climatic yield was relatively stable in 2008–2018. In particular, there were many years of continuous reduction in winter wheat yields prior to 2006. Thereafter, the impact of climate change on winter wheat yields became smaller. This study expands our understanding of the complex interactions between climate variables and crop yield but also provides practical recommendations for enhancing agricultural practices in this region

## 1. Introduction

Winter wheat, one of the main food crops in China, with the largest traditional planting area [1], is highly sensitive to weather conditions [2]; its year-round growth cycle makes it particularly vulnerable to climate change. As winter wheat is affected substantially by varying climates and management practices, it is important to predict its responses to variable climates [3]. It is well known that winter wheat is dormant in winter but very active in spring and summer. Research suggests that climate change might reduce winter wheat yields in major wheat-growing regions worldwide [4]. In addition, with the growing global population, food production is facing increasing pressure. Therefore, it is imperative to conduct further research on the response of winter wheat production to climate change, especially in spatial and temporal terms [5]. These studies are essential for maintaining sustainable development and ensuring food security in the study region, particularly in regions like Shanxi Province, a major winter wheat-producing area in China [6,7]. In light of limited prospects for expanding arable land and China’s continuous population growth, the in-depth investigation of the interplay between winter wheat yield and climatic factors as well as the characteristics of yield changes has far-reaching significance for guaranteeing the security of China’s food production.

Studies of the relationship between climate change and crop yields can be dated back to the early 20th century [8,9,10,11]). Recent research, with the utilisation of climate models based on extensive observational data [12], has revealed that climate change impacts the growth and yield of various crops in complex ways, involving factors like temperature, precipitation, and carbon dioxide concentration [13]. For instance, Gong et al. (2020) [14] calibrated the model DSSAT-CERES-barley and verified it based on agro-climatic information of the Tibetan Plateau. It is interesting that significant fluctuations in production potential could be observed due to climate change over the past 40 years, particularly at lower elevations. However, production potential at high-elevation sites was relatively stable. The integration of information technology and agricultural science has led to advancements like the geographic information system (GIS), which has become crucial in analysing spatial distribution and changes in agricultural production [15]. This provides an important platform for the evaluation of agricultural production development and related decision-making. Li et al. (2021) [16] proposed a crop yield simulation framework by combining a cropping system model (CropSyst) with a geographic information system (GIS). This not only filled the gap between local cropping system models and regional estimates of crop yields but also provided more reliable information for the decision-making process. A comparison of the planted area of olive trees with the actual area by Aydi et al. (2016) [17] illustrated and explained the suitability of the selected area for olive cultivation. Therefore, scientific methods are required for the yield expansion of olive crops. Zhu (2019) [18] analysed a range of factors that exert influence on wheat yield in the mountainous areas of southeast Shandong Province separately, including climatic change, topography, and soil. A hierarchical analysis method was adopted in their study, and different weights were assigned to factors, like river and land use type, that affect wheat planting. Finally, a comprehensive zoning map of wheat planting in Wulian County was compiled with the help of GIS technology. Ding et al. (2022) [19] used the geodetector model to explore the influence of various natural factors, such as precipitation, temperature, vegetation type, soil type, and elevation, and anthropogenic factors, like land use type, population density and GDP, on NDVI changes in the study area.

At present, the main wheat-producing areas in China are the Huang-Huai wheat region and the middle and lower reaches of the Yangtze River [20,21,22,23,24,25]. In general, crop yield fluctuations consist of three parts: yield trend (attributed to production development), climatic yield variation (attributed to changes in average climatic elements and extreme weather events), and random error [26]. Climatic yield variation indicates the changes in productivity levels explained by climatic factors [27]. However, studies focusing on the climatic yield of wheat, especially in Shanxi Province, are relatively rare. In addition, current research in Shanxi predominantly centres on the county scale, with areas like Shouyang County and Jixian County often being the focus [28,29]. However, province-wide assessments of wheat’s climatic yield are less common. Studies that take the whole province as the target area rather than the county to evaluate wheat climatic yield are quite few. Given that Shanxi Province is one of the major winter wheat production areas in China, this study looked into the spatial and temporal distribution characteristics of precipitation, cumulative temperature during the reproductive period of winter wheat, actual yield, and climatic yield in the region. The correlation of factors affecting climatic yield was also investigated to verify an accurate relationship between them. This is critical for the future analysis and study of change patterns for winter wheat climatic yield in the region. This information can assist government departments in effectively managing grain production, identifying key areas for yield improvement, and implementing macro-controls in grain-growing regions. Therefore, the temporal and spatial distribution characteristics of precipitation, accumulated temperature during winter wheat growth periods, and actual yield and climatic yield of winter wheat in this region should be studied. Apart from this, a correlation analysis of the relationship between the influencing factors and climatic yield is also required to help us understand how these variables affect each other. It is of great significance to analyse and study the change law of winter wheat climatic yield in the future, quickly and accurately grasp the yield production plan formulated by government departments, determine the key development areas of winter wheat in the future, and implement macro-control of winter wheat planting areas.

In this study, we first established a database encompassing actual yield, climatic yield, and nine influence factors. Second, we spatially visualised the influence factors, including actual yield, climatic yield, wind speed, sunshine time, cumulative temperature, precipitation, normalised difference vegetation index (NDVI), soil type, elevation, slope, and slope direction. Third, with the help of a GIS, the spatial visualisation of the main influencing factors, i.e., actual yield, climatic yield, wind speed, sunshine hours, accumulated temperature, precipitation, NDVI, soil type, elevation, slope, and slope direction, can make our research more intuitive. Next, variable information of the factors, which was used as input data for calculations, was extracted from the data obtained in the previous step. The use of “Single Factor Detector” and “Interaction Detector” allowed the factors that affected the actual yield and climatic yield to be effectively analysed. By examining day-by-day meteorological data from 12 stations in Shanxi Province between 1964 and 2018, we analysed the spatial and temporal variations of the aforementioned factors during the winter wheat growth period. This comprehensive approach allowed us to understand the spatial-temporal distribution characteristics of climate and their impact on winter wheat yield in Shanxi Province, thereby providing valuable insights for strategic agricultural planning and management in the region.

## 2. Materials and Methods

### 2.1. Description of the Study Area

Shanxi Province (Jin, in abbreviated form) is situated along the middle reaches of the Yellow River, west of the Taihang Mountains, in central China. Spanning 156,000 square kilometres, it is geographically positioned between 34°34′–40°44′ N latitude and 110°14′–114°33′ E longitude. The province borders Hebei to the east, Shanxi to the west, Henan to the south, and Inner Mongolia to the north. Characterised by a temperate, continental climate, Shanxi experiences four distinct seasons with abundant sunlight. The average annual precipitation for a long time period in this area is about 400~650 mm, but the seasonal distribution is uneven. In fact, the precipitation is highly concentrated and heavy in summer from June to August, accounting for 60% of annual records. Temperature ranges vary significantly, with mean annual temperatures between −4 °C and 14 °C. Winter temperatures in the whole province are below 0 °C, while summers, particularly in July, can be quite warm, reaching 21~26 °C. The frost-free period varies across the province, with a longer duration in the south and a shorter duration in the north. For example, the Datong Basin experiences 110 to 140 days, the Wutai Mountain area only about 85 days; the duration for the east mountains and the north of Xinzhou Basin is about 135 to 155 days, and that for the Linfen and Yuncheng basins is about 200 to 220 days. The mean annual frost-free period is shorter than 120 days; thus, wheat can only be planted in one season. The predominant soil type is either chestnut soil (Chinese classification) or Calcic Luvisol (64% sand and 21% clay in the top 0~20 cm and 41% sand and 27% clay in the top 20~120 cm). The groundwater depth is generally below 12 m. Figure 1a,b show the location map and the digital elevation model map (DEM) of Shanxi Province, respectively.

### 2.2. Data Collections

Daily meteorological data from 12 national meteorological stations in Shanxi Province for the period from 1964 to 2018 can be downloaded from the China meteorological data sharing network (http://data.cma.gov.cn/) (accessed on 28 December 2023). These data include variables like the maximum and minimum temperature, precipitation, relative humidity, sunshine hours, wind speed, etc. The distribution of these meteorological stations is illustrated in Figure 1a. Wheat cultivation has a long history in Shanxi Province, with the province’s wheat planting area reaching 503,000 ha in 2023. Data on winter wheat production from 1978 to 2018 and on planting areas and winter wheat production in the counties and cities of Shanxi Province from 2005 to 2018 were obtained from the statistical yearbook of Shanxi Province and China Statistics official website (http://www.stats.gov.cn/) (accessed on 28 December 2023). We also gathered winter wheat production data from 50 counties according to the criteria for delineating winter wheat-planting areas.

### 2.3. Research Methods

#### 2.3.1. Geodetector

The geographical detector method used the factor “q” to measure Spatial Stratified Heterogeneity (SSH) and to make attributions for or by SSH [30,31,32]. It can determine whether there is an interaction between the influence factors, such as various climate factors. The stronger the interaction, the greater the impact of the factor on climatic yield and the closer the relationship between the factor and yield, and vice versa. The examination of variation degree in the spatial distribution of independent variables (e.g., climate factors) and the dependent variable (e.g., climatic yield) helped us to decide whether there was a significant relationship and potential causal mechanism between the independent variable and the dependent variable. If there were significantly difference between the two factors in the attribute mean values, this indicated that the independent variable plays an important role in influencing the spatial variation of the geographic data. Using the q-value measure, the expression was defined as follows:(1)$$q=1-\frac{\sum_{h-1}^{L}Nh\sigma_{h}^{2}}{N\sigma^{2}}=1-\frac{SSW}{SST}$$
(2)$$SSW=\sum_{h-1}^{L}Nh\sigma_{h}^{2},SST=N\sigma^{2}$$
where *h* = 1, …, l is the stratification of variable *Y* or factor *X*; *N_h_* and *N* are the numbers of cells in layer *h* and the whole area, respectively; $\sigma_{h}^{2}$ and σ are the variance of *Y* in layer *h* and the whole region, respectively; and *SSW* is the sum of the intra-stratum variance and *SST* is the total variance of the whole region. A high q-value that ranged from 0 to 1 indicated a stronger explanatory power of an independent variable *X* for the dependent variable *Y*.

An interaction detector, widely used in fields such as social science, medicine, and environmental science, was the analytical tool adopted in this study. Its main function was to assess whether there was an interaction amongst the effects of different independent variables on the dependent variable (Figure 2). Specifically, it assessed whether the effects from factors X_1_ and X_2_ were enhanced or weakened when they acted together on the dependent variable Y or whether the effects of these factors on Y were independent and/or had no interaction. The method of evaluation was as follows:

First, calculate the q-values of the two influencing factors X_1_ and X_2_ on Y separately. Second, calculate the influence on Y when these two factors interact (i.e., the new polygonal distribution formed after the two layers X_1_ and X_2_ overlap each other) and compute their q-values. Third, compare the magnitude of these three q-values. Due to the different results of the comparisons, the interactions amongst factors could be subdivided into the following scenarios.

GIS geodetectors were used to qualitatively analyse the relationship among nine factors in 2018, including precipitation during winter wheat growth periods, accumulated temperature, wind speed, sunshine hours, elevation, slope, slope direction, NDVI, and soil type, and the actual yield and climatic yield in the study area. On the basis of this, the influence degree of different influencing factors on winter wheat climatic yield will be discussed in detail.

#### 2.3.2. Wavelet Analysis

Wavelet analysis, also known as multi-resolution analysis, is adept at reflecting the change cycle of elements across different time scales [33]. In this study, Morlet wavelet analysis was employed to calculate the wavelet coefficients and wavelet variance of meteorological factors in Shanxi Province. MATLAB software ((64bit) 2016a version 9.0) was used to carry out Morlet wavelet analysis.

For a given wavelet function Ψ(*t*), the continuous wavelet in time series *f* (*t*) ∈ L^2^(R) was transformed as follows:(3)$$Wf(a,b)={\left|a\right|}^{-\frac{1}{2}}\int_{-\infty }^{+\infty }f(t)\overset{-}{\Psi }(\frac{t-b}{a})dt$$
where *a* is the scale parameter that reflects the period length of the wavelet, *b* is the paralleling parameter that reflects paralleling over time, $\bar{\Psi }$ is the complex conjugate function of Ψ(*t*), and *W_f_* (*a*, *b*) is the wavelet transform coefficient. Since time series are often discrete in practice, the discrete form of the above equation is often used, which is below:(4)$$Wf(a,b)={\left|a\right|}^{-\frac{1}{2}}\Delta t\sum_{k=1}^{N}f(k\Delta t)\overset{-}{\Psi }(\frac{k\Delta t-b}{a})$$

The graph of the wavelet transform coefficient was a two-dimensional contour graph for *W_f_* (*a*, *b*), with *b* taken as the horizontal axis and *a* taken as the vertical axis. The variation characteristics in the time series could be obtained from the wavelet transform coefficient graph.

Wavelet variance, obtained by integrating the square of the wavelet transform coefficients of a in the time domain, was calculated by the following equation:(5)$$Var(a){\int_{-\infty }^{+\infty }\left|Wf(a,b)\right|}^{2}db$$

#### 2.3.3. Additional Data Sources and Pre-Processing

Microsoft Office Excel 2021 was applied to process the collected data, creating line charts of both rainfall and accumulated temperature during the growth period of winter wheat from 1964 to 2018 and winter wheat yield from 1978 to 2018. Data on historical winter wheat yield (kg·ha^−1^) were directly obtained from the China rural statistical yearbook (1985–2021), the statistical yearbook of Shanxi Province 2022, and the China Statistics official website (http://www.stats.gov.cn/) (accessed on 28 December 2023). In order to improve the accuracy of the data, the statistical data suspected to be wrong were partially corrected on the basis of investigation and a literature review.

Inverse distance weighting (IDW) in ArcGIS 10.8 software was used to interpolate the above meteorological elements and different levels of yield [34,35,36,37]. A 3-year sliding average method was applied to separate the climatic yield [26,38,39,40] and decompose the winter wheat yield into (1) the growth of crop yield advanced by technological progress, agricultural policy, and the growth of material inputs, which reflects the level of socio-economic and technological development of a certain period of history—this is referred to as time-technological trend yield, or trend yield for short; (2) fluctuations in crop yield caused by differences in meteorological condition classes—the corresponding yield component is called climatic yield, reflecting the impact of meteorological fluctuations on yield; and (3) the yield error value, which is generally affected by test conditions and environmental factors, such as temperature, humidity, air pressure, etc. The formula is as follows:(6)$$Y=Y_{\omega }+Y_{t}+\varepsilon$$
where *Y* is the actual crop yield, *Y_ω_* is the trend yield, *Y_t_* is the climatic yield, and *ε* is the yield error influenced by random factors.

A coefficient of variation (*CV*) analysis was employed to study the variation magnitude in climatic yields across the region. The *CV* is the ratio of the mean square deviation to the mean, reflecting the degree of dispersion of different observation series. The formula for calculating the *CV* is given as follows:(7)$$CV=\frac{\sqrt{\frac{1}{n}\sum_{i=1}^{n}{\left(x_{i}-\bar{x}\right)}^{2}}}{\bar{x}}$$
where $x_{i}$ is the value of the element in year i and $\bar{x}$ is the multi-year average value.

## 3. Results

### 3.1. Spatial and Temporal Distribution Characteristics of the Precipitation and Accumulated Temperature

#### 3.1.1. Spatial Distribution Characteristics

Figure 3a illustrates the average precipitation during the growth period of winter wheat from October to June in Shanxi Province in 1964–2018. Based on the graphs, one can see that the precipitation during the growth period of winter wheat in the last 55 years shows a decreasing trend from south-east to north-west, with precipitation ranging from 120 to 191 mm. It can be seen that higher precipitation was observed in Yuncheng City, the most parts of Yangquan City and some counties of Changzhi and Datong City, with precipitation ranging from 177 to 191 mm. Areas like Jiexiu City, Lingshi County, Zuoyun County, and Shanyin County in the south-central and northwestern parts of Shanxi Province experienced less than 148 mm during the growth period of winter wheat. In contrast, the total precipitation during the growth period of winter wheat was relatively high in the eastern and southern regions. The average value of the total precipitation in the study area was 157.45 mm, which was far below the water requirement of winter wheat. Therefore, irrigation with a certain amount of water supply is necessary to ensure a high yield of winter wheat.

As can be seen from Figure 3b, the accumulated temperature during the growth period of winter wheat gradually increased from northeast to southwest, ranging from 464 to 2282 °C. The lowest values were noted in Wutai County, Lingqiu County, and other areas, while the highest was recorded in the southwest, ranging from 1985.52 to 2282.64 °C. Impacted by factors like topography and climate, significant differences in the accumulated temperatures in different areas of Shanxi Province can be clearly observed. For example, accumulated temperature in the northern mountainous and plateau areas was low, while values for the southern areas of the Yellow River and Taihang Mountains were relatively high. Therefore, it is necessary to choose the right varieties and planting periods in line with the local climate and land conditions when winter wheat is planted. This would certainly ensure growth and yield.

The wind during the growth period of winter wheat speed increased from southeast to northwest, as depicted in Figure 3c. The wind speed ranged from 440 to 616 m s^−1^, with the lowest values occurring in most counties of Linfen City and in the northwestern counties of Changzhi City (438.69–472.22 m s^−1^). In contrast, the highest wind speeds were observed in the northwest, ranging from 567.11 to 609.08 m s^−1^. This pattern correlates with the terrain, with higher and windier areas in the northern and western parts of Shanxi Province, while the southern and eastern areas, being flatter, experienced lower wind speeds.

As shown in Figure 3d, the sunshine hours during the growth period of winter wheat gradually increased from southwest to northeast, ranging from 1382 to 1863 h. The least sunshine hours appeared in most counties of Yuncheng city and some counties of Linfen city, ranging from 1382 to 1499 h. In comparison, Shuozhou city and its surrounding counties had the longest sunshine durations. The average sunshine duration during the growth period of winter wheat for the entire study area was 1591.51 h.

#### 3.1.2. Temporal Distribution Characteristics

The inter-annual variation of mean annual precipitation and accumulated temperature during the growth period of winter wheat in Shanxi Province from 1964 to 2018 is shown in Figure 4. The annual mean precipitation during the growth period of winter wheat was 155.13 mm in the study area, showing a slight increasing trend at a rate of 0.1299 mm a^−1^ (Figure 4a). The maximum precipitation was recorded in 1964 (242.44 mm) and the minimum was in 2000 (91.27 mm). The annual mean accumulated temperature during the growth period of winter wheat gradually increased at a rate of 6.7688 a^−1^, peaking in 2007 (1930.24 °C) and reaching its lowest in 1976 (1437.73 °C); see this in Figure 4b.

#### 3.1.3. Wavelet Analysis of Periodic Variation

The wavelet analysis revealed distinct cyclic oscillations in both annual mean precipitation and accumulated temperature during the growth period of winter wheat in Shanxi Province from 1964 to 2018. The variogram of the wavelet analysis reveals the distribution of the fluctuation of energy of the annual mean precipitation and annual mean accumulated temperature during the growth period on a time scale. As shown in Figure 4c, there were five obvious peaks of annual mean precipitation on a long time scale during the growth period, corresponding to 6, 10, 23, 42, and 56 years, respectively. Among the five cycles, the 42-year cycle wavelet variance was the largest. The results indicate that the oscillation reached its peak value at around 42 years, which was the first main cycle of annual mean precipitation during the growth period of winter wheat. Figure 4d shows that the obvious peaks of the annual mean accumulated temperature during the growth period were mainly located at the 56-year mark. The peak difference during this period is self-evident, indicating that the periodic oscillation was the strongest around 56 a, which was the first major cycle of the annual mean accumulated temperature. On a time scale of 1–36 years, the peak changes at 12 and 26 years were insignificant and negligible. Therefore, the 36-year time scale was the second major cycle.

The annual mean precipitation had five distinct “negative-positive” alternating oscillatory cycles on a time scale of about 42 a. This indicates that Shanxi Province has experienced an alternation process of “dry-abundance” in the last 55 years, with more intensive oscillation cycles on a time scale of 10~20 a (Figure 4e). Annual mean accumulated temperature during the growth period oscillated on a time scale of 50~60 a (Figure 4f).

### 3.2. Spatial and Temporal Distribution Characteristics of the Actual Yield and Climatic Yield

#### 3.2.1. Spatial and Temporal Distribution Characteristics of the Actual Yield

The actual output from 50 continuous cropping wheat counties across 11 cities in Shanxi Province from 2008 to 2018 is shown in Figure 5. Noticeably, the spatial distribution trend shows higher yields in the southwest and lower yields in the northeast. High-production areas of winter wheat in the study area were mainly located in Changzhi City, Jincheng City, the eastern part of Yuncheng City, and some counties in Xinzhou City and Linfen City, while the low-production areas were mainly located in the counties of Lvliang City and Shuozhou City.

It also can be seen from Figure 1 and Figure 3 that the spatial distribution pattern of actual winter wheat yield in Shanxi Province was closer to the distribution of precipitation, elevation, and wind speed compared with accumulated temperature and sunshine hours.

Figure 6 presents the trend of actual winter wheat yields in each district of the entire study area in 1978–2018. As can be seen from the figure, the average actual yield in the district shows a significant increasing trend of 47.827 kg ha^−1^yr^−1^.

The common fitting methods usually used for separating trend returns of yields include linear, exponential, polynomial, and logarithmic functions. Different fitting methods are suitable for different data patterns and contexts, and even the same fitting method could cause dramatically different results for different time periods and/or regions. This might be related to the characteristics of the data itself, differences in data quality, or changes in environmental, economic, or policy factors. Therefore, it is important to choose the appropriate method to separate the trend yields from the climatic yields based on the crop production situation and climatic characteristics. Different regions might have different natural characteristics, climatic conditions, economic backgrounds, production management styles of individual farmers, and agricultural technologies. Therefore, factors that affect crop yield must be taken into account when the crop yield in different regions is being analysed. A representative, applicable, and reasonable separation method is also required. Only appropriate methods can ensure the accuracy of the assessment of the climate change impact on climatic yield. Ultimately, the improvement of agricultural production quality and efficiency is beneficial for promoting the process of rural modernisation.

In this study, we analysed the winter wheat of Shanxi Province using historical yield data from 1978 to 2018. In previous studies, different methods were used to separate the trend yield and climatic yield from the actual yield. The 3-year sliding average method, taking the 3-year average value of time series data, can eliminate seasonal fluctuations in the data and give a more realistic long-term trend. In fact, variables may not always be linearly correlated. Curve fitting is conducted to select a suitable curve type to fit the observed data, and the cubic polynomial method is utilised to analyse the relationship between two variables by fitting a cubic polynomial curve. Both the 3-year sliding average method (Figure 7a) and the cubic polynomial method (Figure 7b) were employed to fit the trend yield of winter wheat. The applicability and reasonableness of the above two methods have been evaluated and compared with the actual development in current society. In this study, a more suitable method, which could provide reliable support and references for agricultural production, was chosen to separate the climatic yield component of winter wheat yield. The fitting results of the trend yield are shown below.

Although the trend yields fitted by both methods were very similar to the trend of increasing yields due to technological advances in social production, our results show that the trend yields fitted using cubic polynomials were less consistent with the current state of social production in 1990–2005. Since the reform and opening up, and with the rapid economic development, the incentives of rural production have increased unprecedentedly, indicating that agricultural production was expected to recover and develop rapidly. However, a prediction study using a cubic polynomial function shows that the growth rate of trend production gradually slowed down from around 1990 and only began to accelerate again in 2005. Obviously, this development trend is not coordinated with the current situation of social production. That is to say that this method was not suitable for our study and a better choice needed to be considered. In contrast, the 3-year sliding average method aligned more closely with the actual increase or decrease changes in real production, leading us to select this method to simulate trend production.

#### 3.2.2. Spatial Distribution Characteristics of the Climatic Yield of Winter Wheat in Shanxi Province

Figure 8 presents the distribution and variation coefficient of climatic yield in 50 counties of Shanxi Province from 2008 to 2018. In general, the variation coefficient of winter wheat climatic yield in Shanxi Province was relatively stable, with fluctuations between 1.45% and 4.5%. The variation coefficient fluctuated sharply in the southwestern part of the study area (e.g., Shilou County in Lvliang City), exceeding 3.91%. Conversely, in counties and districts such as Changzhi and Qinshui, where precipitation was more abundant and normal conditions of accumulated temperature were stable during the growth period of winter wheat, the variation coefficients were smaller, ranging from 1.45% to 2.08%.

As can be seen from Figure 1 and Figure 3, the spatial distribution pattern of the climatic yield of winter wheat in Shanxi Province is closer to the distribution of precipitation, sunshine hours, and accumulated temperature compared with elevation and wind speeds.

#### 3.2.3. Temporal Distribution Characteristics of the Climatic Yield of Winter Wheat in Shanxi Province

As can be seen from Figure 9, the years prior to 2006 experienced more significant declines in winter wheat yield due to climate change. Since 2006, the frequency of years with negative climatic yields has been decreasing, with both the number and magnitude of their impacts being smaller than in previous periods. In addition, the fluctuations in yields are relatively small after 2006.

### 3.3. Correlation Analysis of Factors Affecting the Yield of Winter Wheat

#### 3.3.1. Single-Factor Detection by Geodetectors

Table 1 shows the results of single factor analysis for the actual winter wheat yield and climatic yield in the study area in 2018, with a range of meteorological factors. It can be seen that sunshine duration, precipitation, wind speed, and accumulated temperature had relatively high impacts, while slope direction and gradient had lower impacts.

#### 3.3.2. Multi-Factor Detection by Geodetectors

In the multi-factor interaction analysis based on Geodetectors, it was found that there were significant correlations among most factors, and the comprehensive influence of most of them on yield was greater than that of individual factors. Figure 10 indicates that the combined impact of two factors on yield was generally greater than their individual impact, suggesting a two-factor enhancement. For example, the correlation analysis for the actual yield and each meteorological factor was as follows: precipitation ∩ accumulated temperature = 0.469, precipitation ∩ elevation = 0.437, precipitation ∩ wind speeds = 0.433. With regard to the climatic yield, the correlation analysis for the climatic yield and each meteorological factor was as follows: precipitation ∩ accumulated temperature = 0.376, precipitation ∩ sunshine duration = 0.35, cumulative temperature ∩ sunshine duration = 0.338. The results suggest that the nine meteorological factors (as shown in Table 1) interacted with each other and there was mutual reinforcement among them. It also shows that they had a strong mutual enhancement effect on yield. Overall, their individual influence became even greater. This was particularly true for the precipitation and accumulated temperature, which were the main limiting factors during the winter wheat growth periods.

## 4. Discussion

### 4.1. Spatial-Temporal Variations of Precipitation and Accumulated Temperature during the Winter Wheat Growth Period

The spatial distribution of precipitation during the winter wheat growth period in Shanxi Province shows a general increasing trend from northwest to southeast. This finding is consistent with the article by Gong (2022) [41]. The areas with high precipitation were mainly concentrated in the south and east of the study area, including most of the counties in Yuncheng and Yangquan and a few counties in Changzhi and Datong [42]. Shanxi Province, which is geographically between the Loess Plateau and the area north of the Yangtze River, is located in a typical climate transition zone. It is significantly influenced by the atmospheric circulation and air pressure distribution. Also, more influences come from the continental climate, as the area is deeply located in the continent [43,44]. In addition, the unevenness of terrain and complexity of topography, as well as high mountains, mountain ranges, and mountain basins, all impose impacts on the direction and intensity of airflow [45]. As a result, precipitation is unevenly distributed in space during winter wheat growth periods. In terms of spatial distribution characteristics, spatial variations and variation patterns between the accumulated temperature in Shanxi Province and that of China are different [46]. The mean value of accumulated temperature during the winter wheat growth period in Shanxi Province in 1964–2018 inherited a latitudinal spatial distribution characteristic, with higher temperatures in the south and lower temperatures in the north, differing from the meridional distribution pattern observed across the province. The topography of the country is high in the west and low in the east, with a three-tiered, stepped distribution. The strong topographic elevation difference enables the temperature to change along the contour line in the east–west direction [47]. However, the topographical features of Shanxi are not obvious in comparison with those of the whole nation. The average annual accumulated temperature is greatly influenced by the latitude factor. The accumulated temperature gradually decreases with the increasing latitude. The Taihang Mountains and the Lvliang Mountains on the east and west sides of Shanxi and the five major basins run through the whole central part from northwest to southeast, so the temperature contour shows a convex variation. The southwestern part with the highest temperature is located right in the Yuncheng Basin. This is in agreement with findings from Du et al. (2021) [48]. In this study, the periodicity of precipitation and accumulated temperature during winter wheat growth periods in Shanxi Province was derived from the wavelet analysis, and our research methods are consistent with those of Shao et al. (2023) [49]. On a time scale of nearly 42 years, which was the first big cycle of annual mean precipitation during the growth period, the regularity of mean annual precipitation in Shanxi Province shows the strongest oscillation. The strongest oscillation of accumulated temperature occurred on a cycle of 56 years, which was the first major cycle of annual mean accumulated temperature.

### 4.2. Impacts of Climate Change on Winter Wheat Yield in Shanxi

In this study, we found that the actual yield of winter wheat in Shanxi Province exhibited a significant increasing trend, with 47.827 kg ha^−1^ yr^−1^. This is consistent with the findings of Zhong et al. (2022) [50]. However, the results of our study are of more general significance in comparison with those presented by Zhao, who focused on the two cities of Linfen and Yuncheng. The climatic yield of winter wheat was closely related to the precipitation and accumulated temperature during the winter wheat growth period. The amount of precipitation was far less than the amount of water required for winter wheat. Appropriate irrigation must be carried out in a bid to ensure the high yield of winter wheat [51,52,53]. This is consistent with findings reported by Ning et al. (2020) [54], but slightly differed from those presented by Li and Hu (1993) [55]. In their studies, yield was not strongly and inversely correlated with temperature, and the meteorological yields in Shanxi were highly variable and closely correlated with precipitation. In particular, yields in years with high precipitation tended to be high. On the contrary, yields in most of the early years were low, stemming from the differences in the research methods and data selected. Thanks to the improvement of research methods and techniques, the results now become more accurate than before. In some of the years in the period 1964–2018, the accumulated temperature during winter wheat growth periods in Shanxi Province showed significant fluctuations, causing great impacts on the growth and development of winter wheat. For example, years with either low or high temperatures could result in slow or stagnant growth. These conditions make the growth and development of winter wheat unpredictable. So, a proper planting strategy must be designed based on planning. Due to the high frequency of extreme weather phenomena in recent years, the occurrence of catastrophic weather such as heavy rains, droughts, floods, and storms is on the rise and will lead to severe impacts on agriculture in the future [56,57,58]. Therefore, more attention should be paid to how often extreme weather occurs so that better countermeasures can be developed [59,60].

### 4.3. The Synergistic Effect of Multiple Meteorological Factors on Winter Wheat Yield Should Be Taken into Account

The study and formulation of agricultural measures and strategies for future agricultural production are imperative to well accommodate the process of climate change. The aim of this study is to put forward appropriate suggestions on this issue based on the analysis of a large amount of long-term data. However, some shortcomings and limitations existed in the impact factor analysis. First, only a qualitative analysis of the relationship between yield and each impact factor using geodetectors for the impact factor analysis was conducted; there was a lack of further construction of mathematical models for quantitative calculations [61,62,63]. How to quantify the qualitative factors so as to improve the accuracy of the results and conduct predictive simulations is the direction that we should explore in the future. In addition to this, there are many other climatic factors that have non-negligible effects on the yield of winter wheat besides those selected in this study. Shanxi Province is located in an arid and semi-arid climatic region [28,41] where food production is greatly affected by climate; therefore, this study mainly focused on the impact of meteorological factors on the yield of winter wheat. In the future, we will look into the synergistic effects of multiple climate factors on winter wheat yields. Finally, extreme weather is also a cause of fluctuations in winter wheat yields. The next step is to study the frequency of extreme weather and its typical situation in Shanxi Province. This will give a better understanding of the impact of climate change and disasters on agriculture and provide a scientific basis for improving winter wheat production together with guaranteeing food security in general. The results of the previous studies are helpful to formulate agricultural strategies so that food production can be improved and the sustainable development of agriculture can be promoted.

## 5. Conclusions

This study mainly focused on the impacts of meteorological factors on winter wheat yields in Shanxi Province. We analysed the spatio-temporal variation characteristics of nine meteorological factors with the help of a GIS. The precipitation during the winter wheat growth period in the study area showed an increasing trend from northwest to southeast, while the mean accumulated temperature had zonal spatial distribution characteristics of high in the south and low in the north. This certainly affects the spatial distribution pattern of winter wheat yields in Shanxi Province. The actual and climatic yields of winter wheat were comprehensively evaluated on a spatio-temporal scale. In addition, we looked into the correlation analysis between meteorological factors and winter wheat yield. The actual winter wheat yield was significantly correlated with the precipitation and the accumulated temperature during the growth periods. The variation coefficient of winter wheat climatic yield was relatively stable. In particular, climate change imposed a significant impact on winter wheat yield, and there were many years of continuous reduction prior to 2006. Thereafter, there has been a gradual decrease in years with negative climatic yields, and the impact of climate change on winter wheat yield has also become smaller than before 2006. Moreover, significant fluctuations in potential production were observed due to climate change over the past 40 years using the wavelet analysis method. The findings presented here expand our understanding of the complex interactions between climate variables and crop yield that have the potential to mitigate the adverse effects of climate change on winter wheat production. The insights garnered from this study are crucial for implementing targeted interventions and sustainable agricultural practices in the study region.

## Figures and Tables

**Figure 1 plants-13-00706-f001:**
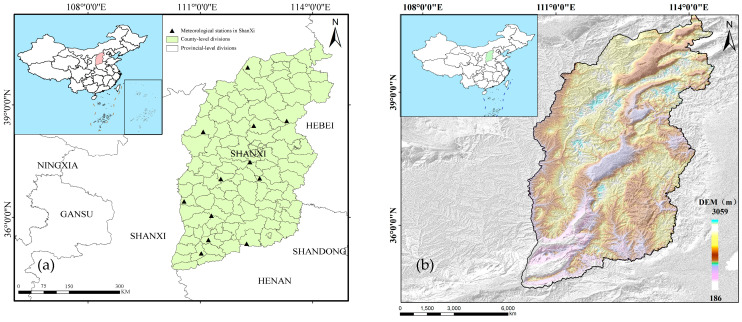
Distribution of the meteorological stations (**a**) and the digital elevation model map (DEM) (**b**) of Shanxi Province in China.

**Figure 2 plants-13-00706-f002:**
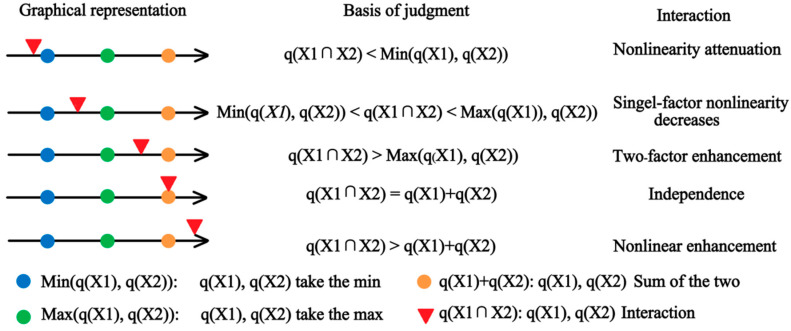
Geodetector interaction analysis results.

**Figure 3 plants-13-00706-f003:**
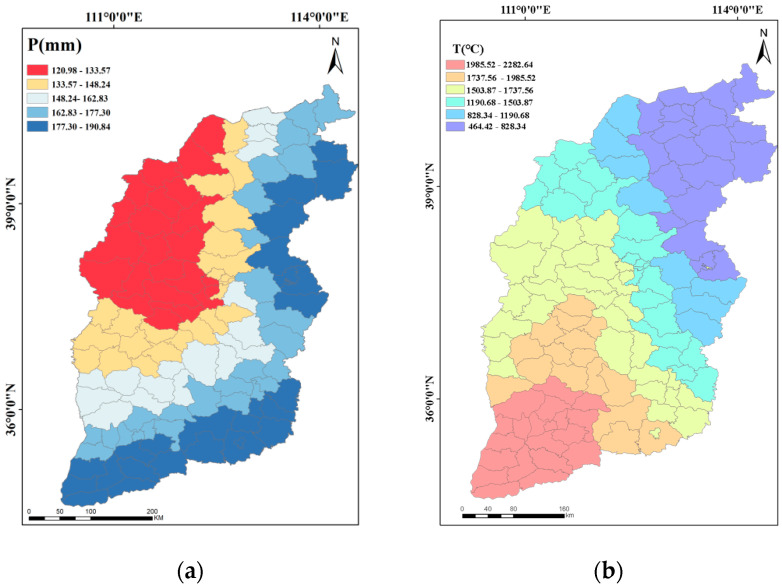
Spatial distribution of precipitation (**a**), accumulated temperature (**b**), wind speed (**c**), and sunshine duration (**d**) during the growth period of winter wheat (October–June) in Shanxi Province, China.

**Figure 4 plants-13-00706-f004:**
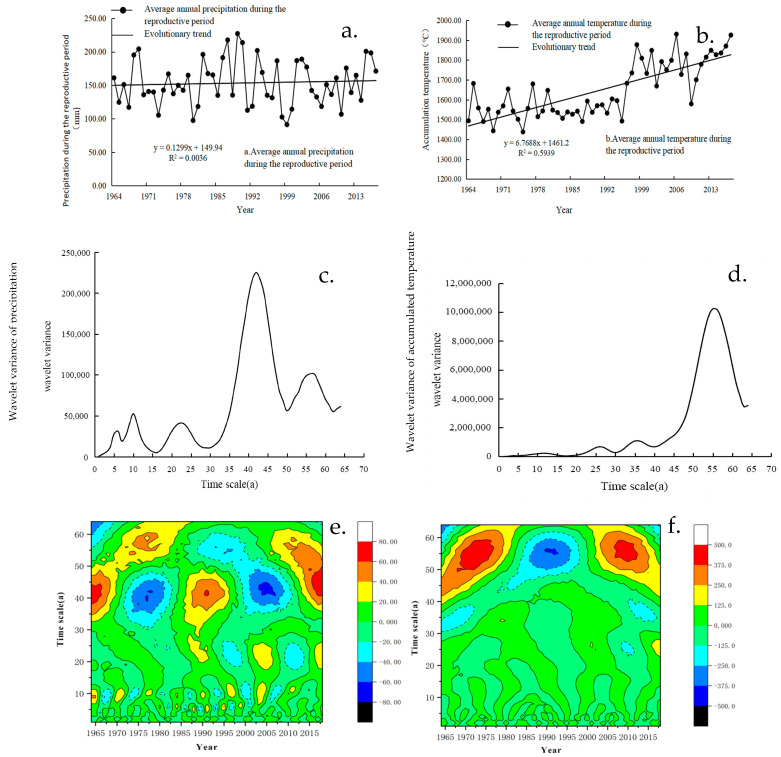
Inter-annual variation of mean annual precipitation (**a**) and mean annual accumulated temperature (**b**), wavelet variance change curves of mean annual precipitation (**c**) and mean annual accumulated temperature (**d**), Morlet wavelet coefficient contours and wavelet variance change curves of mean annual precipitation (**e**), and mean annual accumulated temperature (**f**) during winter wheat growth period in Shanxi Province, 1964–2018.

**Figure 5 plants-13-00706-f005:**
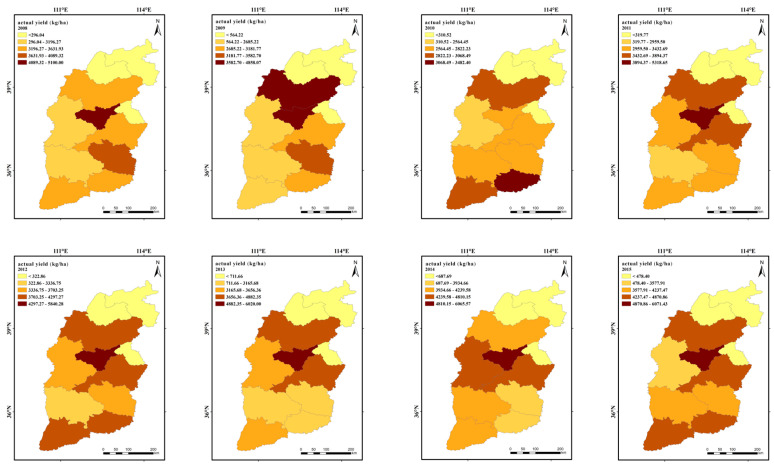
Spatial distribution characteristics of actual winter wheat yield in 50 counties of Shanxi Province, 2008–2018.

**Figure 6 plants-13-00706-f006:**
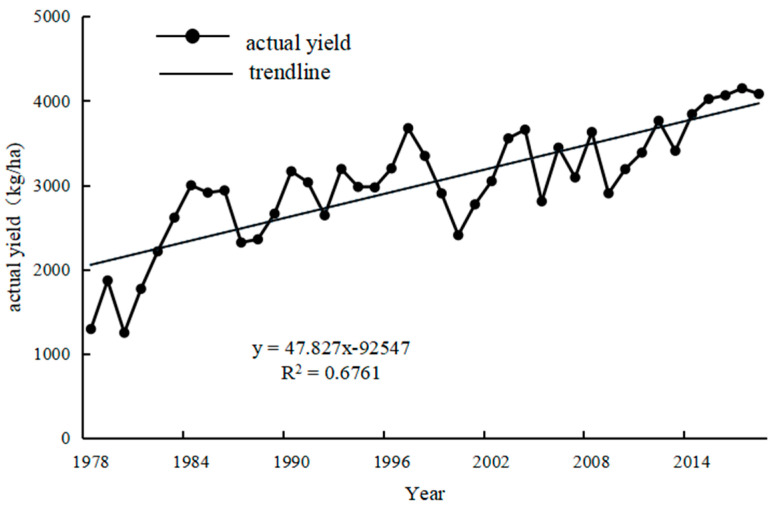
Temporal changes in actual winter wheat production in Shanxi Province, 1978–2018.

**Figure 7 plants-13-00706-f007:**
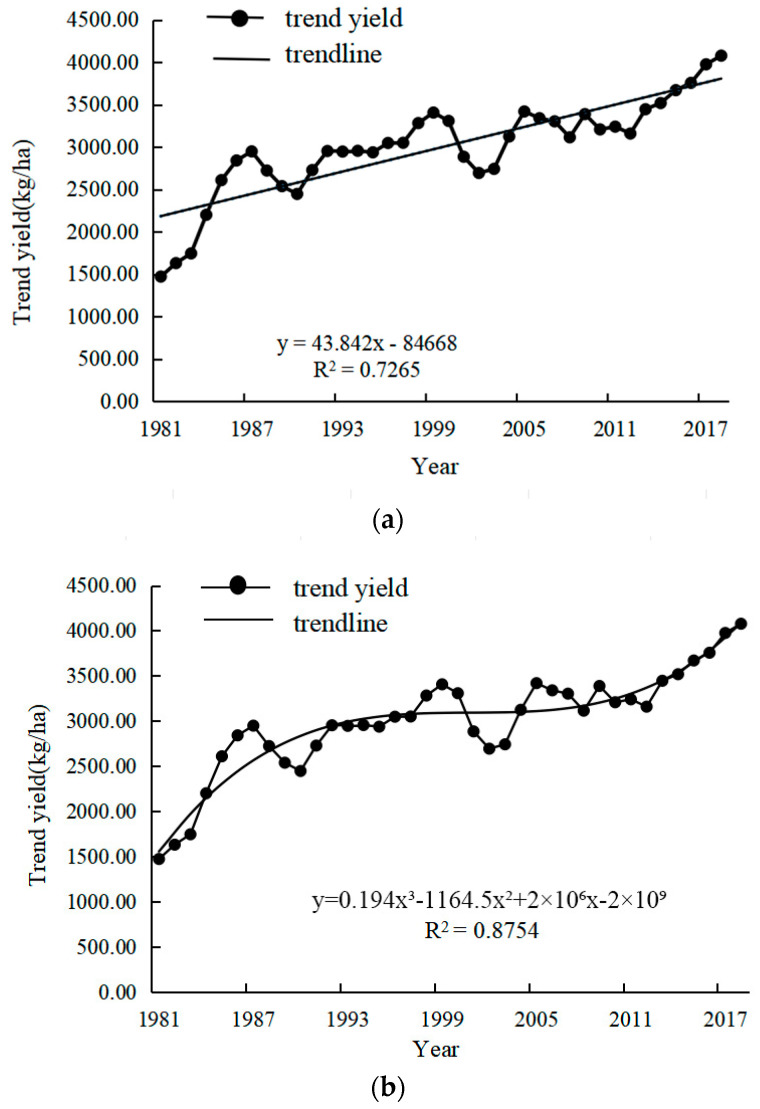
The trend of historical winter wheat yield fitted by 3-yr moving average method (**a**) and cubic polynomial method (**b**).

**Figure 8 plants-13-00706-f008:**
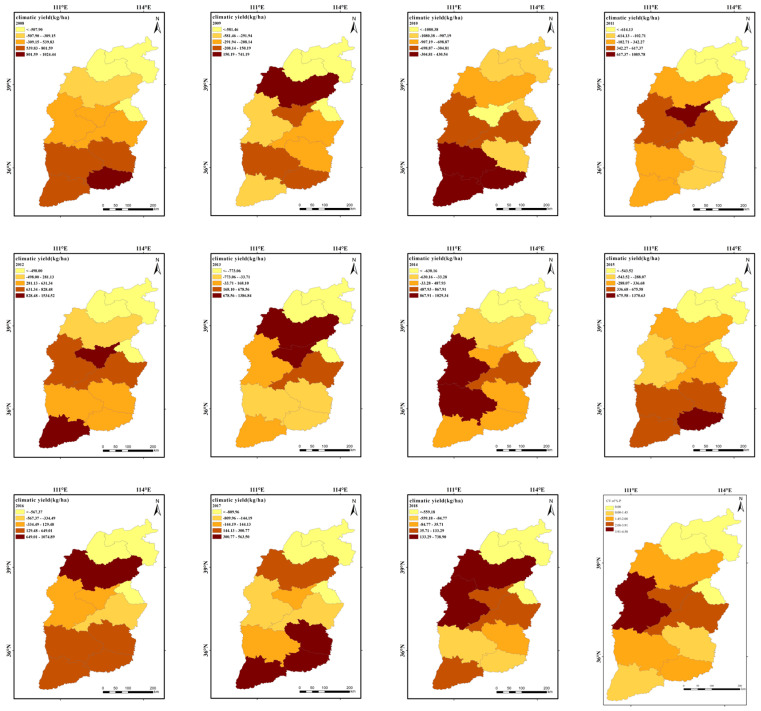
Spatial distribution of the climatic yield and the coefficient of variation in 50 counties of Shanxi Province, 2008–2018.

**Figure 9 plants-13-00706-f009:**
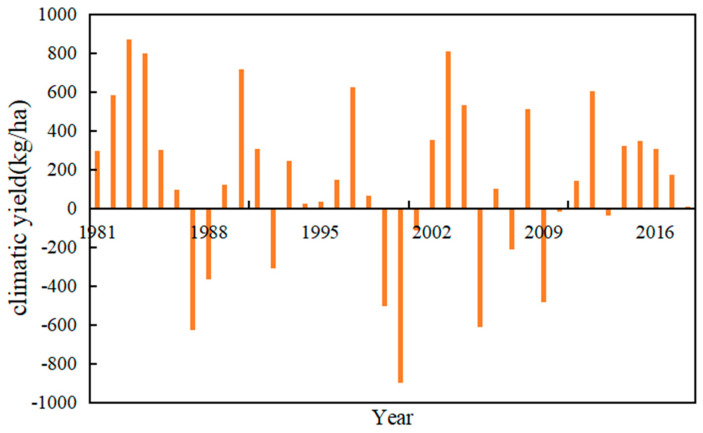
Multi−year climatic yield maps.

**Figure 10 plants-13-00706-f010:**
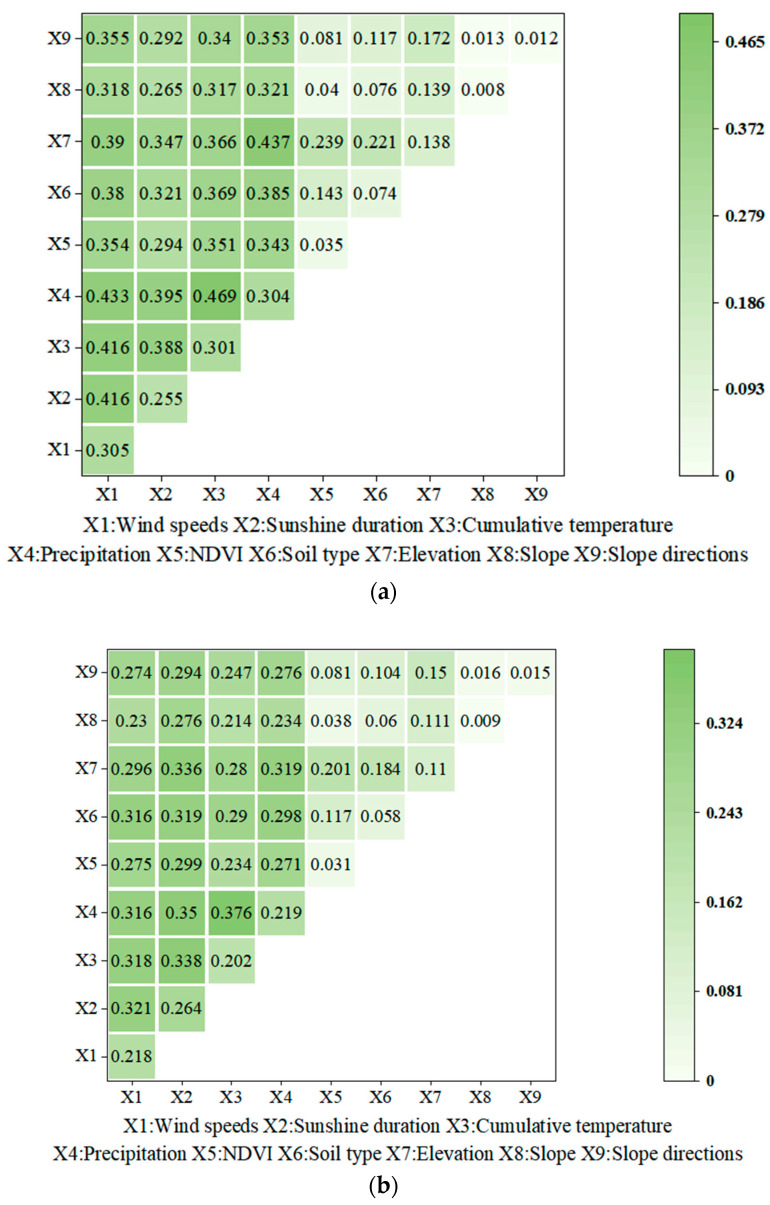
Interaction detection results of factors affecting the actual yield (**a**) and climatic yield (**b**) during the growth periods.

**Table 1 plants-13-00706-t001:** Single-factor analysis of winter wheat yield influencing factors in Shanxi Province.

Factor	Wind Speed	Sunshine Hours	Accumulated Temperature	Precipitation	NDVI	Soil Type	Elevation	Slope	Slope Direction
Actual yield (q)	0.3048	0.254	0.301	0.3041	0.034	0.074	0.138	0.008	0.012
Climatic yield (q)	0.217	0.264	0.201	0.219	0.031	0.058	0.109	0.008	0.014

## Data Availability

Data are contained within the article.

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
