# Peer review of "Analysis of the Spatial-Temporal Distribution Characteristics of Climate and Its Impact on Winter Wheat Production in Shanxi Province, China, 1964–2018"

_plants, 2024, doi:10.3390/plants13050706_

Round 1

Reviewer 1 Report

Comments and Suggestions for Authors

The authors have implemented a prototype and comprehensive analysis on weather data affecting winter wheat yield in the province of Shanxi China. The manuscript follows a clear structure with instructive methodology and an integrated interpretation of the results.  I have only some minor comments, mainly for grammatical errors.

Please describe the spatial distribution on yield data and how is this associated with weather data.

Line 89. Please correct : “2010)). However,”

Line 114: Please correct: “can make our esearch more”

Line 196” Please correct: "Singel”

Line 226: Please refer correctly “software EXCEL was” . “The Microsoft Excel software was”

Line 282: Please correct: “In contrast, the highest ind speeds”

Line 306: Please delete the dot. “an annual accumulated. Temperature”

Line 315: Please describe in the materials and methods what model was used for variogram (eg linear, exponential etc)

Line 463: Please correct: “observed nationwide .”

Line 495: Please change: “In particular, harvest in years” to: “In particular, yield in years”

Author Response

We appreciate all of your suggestions and thanks for your constructive comments.

Following your suggestion, as shown in the tracked version, we have significantly improved and revised our full manuscript thoroughly.

Reviewer 2 Report

Comments and Suggestions for Authors

This article significantly contributes to the existing body of knowledge in the field. The findings presented herein offer valuable insights that have the potential to mitigate the adverse effects of climate change on winter wheat production. By shedding light on various aspects of planting management specific to winter wheat cultivation in Shanxi Province, the research not only expands our understanding of the complex interactions between climate variables and crop yield but also provides practical recommendations for enhancing agricultural practices in this region.

The insights garnered from this study can be instrumental in developing adaptive strategies for farmers and policymakers to cope with the challenges posed by changing climatic conditions. Understanding the specific responses of winter wheat to climate variations in Shanxi Province is crucial for implementing targeted interventions and sustainable agricultural practices. Moreover, the research outcomes have practical implications for refining planting management techniques, optimizing resource allocation, and maximizing the overall efficiency of winter wheat cultivation in the face of evolving climatic patterns.

In essence, the results presented in this article go beyond theoretical contributions, offering actionable information that can be directly applied in the agricultural sector. This knowledge is particularly relevant for stakeholders involved in the planning and decision-making processes related to winter wheat cultivation in Shanxi Province, ultimately contributing to the resilience and adaptability of the agricultural system in the context of a changing climate.

Author Response

We appreciate all of your suggestions and thanks for your constructive comments.

Reviewer 3 Report

Comments and Suggestions for Authors

The present document evaluates the impact of climate change on wheat yield in a China's region, present a novel method to study the evolution of yield dissociating the yield evolution related to practices and the yield evolution related to climate in a spatial approach.

The introduction is comple and present the research questions clearly: Only one reference should be changed.

The material and methods are globally are well presented, but you must include where you have obtained the yield data: the accuracy of those data and the variability of those yields: see more details on the document attached.

The results are globally ok, clear, but there are some minor format details to be corrected in the figures.

The discussion is complete and well referrenced.

The conclusions should be reviewed (and in consecuence also in the abstract) as you mainly made a resume of the results.

It's not practical to cite the graphical abstracts in the text and you do not show the figures on the main text, I recommend to include them in the text.

Other minor details in the attached document

Comments on the Quality of English Language

I have found some minor errors at my english level, needs to be checked

Author Response

Thank you very much for taking the time to review this manuscript. We appreciate all of your suggestions and thanks for your constructive comments. Following your suggestion, as shown in the tracked version, we have significantly improved and revised our full manuscript thoroughly. Please see the attachment.

Round 2

Reviewer 3 Report

Comments and Suggestions for Authors

The requested changed have been done, with a special attention given to improve the conclusions and the abtract.

Only two errors in line 588 'is' has to be changed by 'are' nad in line 589 'study' by 'studied'